# Capture of heavy hydrogen isotopes in a metal-organic framework with active Cu(I) sites

I. Weinrauch[1,*], I. Savchenko[2,*], D. Denysenko[3], S.M. Souliou[4], H.-H. Kim[4], M. Le Tacon[4], L.L. Daemen[5], Y. Cheng[5], A. Mavrandonakis[2], A.J. Ramirez-Cuesta[5], D. Volkmer[3], G. Schütz[1], M. Hirscher[1] & T. Heine[2,6]

The production of pure deuterium and the removal of tritium from nuclear waste are the key challenges in separation of light isotopes. Presently, the technological methods are extremely energy- and cost-intensive. Here we report the capture of heavy hydrogen isotopes from hydrogen gas by selective adsorption at Cu(I) sites in a metal-organic framework. At the strongly binding Cu(I) sites ($32\,kJ\,mol^{-1}$) nuclear quantum effects result in higher adsorption enthalpies of heavier isotopes. The capture mechanism takes place most efficiently at temperatures above 80 K, when an isotope exchange allows the preferential adsorption of heavy isotopologues from the gas phase. Large difference in adsorption enthalpy of $2.5\,kJ\,mol^{-1}$ between $D_2$ and $H_2$ results in $D_2$-over-$H_2$ selectivity of 11 at 100 K, to the best of our knowledge the largest value known to date. Combination of thermal desorption spectroscopy, Raman measurements, inelastic neutron scattering and first principles calculations for $H_2/D_2$ mixtures allows the prediction of selectivities for tritium-containing isotopologues.

[1] Max Planck Institute for Intelligent Systems, Heisenbergstr. 3, 70569 Stuttgart, Germany. [2] Jacobs University, School of Engineering and Science, Campus Ring 1, 28759 Bremen, Germany. [3] Augsburg University, Institute of Physics, Universitätsstr. 1, 86159 Augsburg, Germany. [4] Max Planck Institute for Solid State Research, Heisenbergstr. 1, 70569 Stuttgart, Germany. [5] Oak Ridge National Laboratory, Spallation Neutron Source, PO Box 2008, MS6475, Oak Ridge, TN 37831-6471, USA. [6] Wilhelm-Ostwald-Institute of Physical and Theoretical Chemistry, Leipzig University, Linnéstr. 2, 04103 Leipzig, Germany. * These authors contributed equally to this work. Correspondence and requests for materials should be addressed to T.H. (email: thomas.heine@uni-leipzig.de).

Many applications require hydrogen isotopes beyond protium ($^1$H = H): deuterium ($^2$H = D), and also tritium ($^3$H = T), are used as tracers in molecules allowing the atomistic understanding of chemical reaction mechanisms[1] and biological processes[2], deuterated solvents are used in proton NMR spectroscopy, and heavy water is needed as moderator in nuclear reactors. Tritium and deuterium will most likely become fuel components for nuclear fusion reactors, and with the recent breakthrough in the design of the stellarator Wendelstein-7X[3,4] there is re-emerging hope for realizing nuclear fusion as future energy supply. Tritium decays to a precious nucleus, $^3$He, which is required to achieve mK scale temperatures allowing for low-temperature physics. On the other hand, tritium contaminated heavy water is a major fraction of nuclear waste, and its removal is beneficial from environmental viewpoint.

Although hydrogen has the largest isotopal mass differences of all elements, separation remains a challenging task. The standard process for heavy water production is the Girdler sulfide process, which is energy-intensive and offers only small separation factors[5]. Methods with higher separation factors would be more energy efficient, but more importantly they would allow the production of pure deuterium and the separation of radioactive tritium from heavy water deposits.

Recently, two-dimensional crystals, graphene and hexagonal boron nitride, have been suggested as membranes for H/D separation, showing a room temperature H-over-D selectivity of 10 (ref. 6). However, two-dimensional crystals favour the diffusion of H, the much more abundant isotope, while for technological applications materials are required that capture heavier isotopes from low-concentration mixtures. This can be realized by exploiting the stronger adsorption enthalpy of heavier isotopes, as has already been reported for the metal-organic framework (MOF) CPO-27(Co) (also called Co-MOF-74)[7]. CPO-27(Co) shows a selectivity of ∼10 at 77 K based on chemical affinity quantum sieving. Similar results have been obtained by FitzGerald et al. in M-MOF-74 (M = Fe, Co, Ni)[8–10].

According to our previous work[11], further enhancing the adsorption enthalpy of local sites will have a beneficial effect on hydrogen isotope separation: first, it will show even higher separation factors, and it will shift the adsorption/desorption process window to higher temperature. In this vein, Cu(I)-MFU-4l is an interesting MOF candidate: in the parent cage, MFU-4l[12], some of the terminal Zn-Cl sites at the connectors are replaced by Cu(I) metal sites (Fig. 1). Cu(I)-MFU-4l shows remarkably strong adsorption enthalpies for various gases, including H$_2$ (ref. 13).

Here, we confirm our expected performance of Cu(I)-MFU-4l for hydrogen isotope separation. The D$_2$-over-H$_2$ separation factor of 11 at 100 K is achieved as at this temperature the adsorption is thermodynamically controlled due to an isotope exchange effect, where D$_2$ from the gas phase replaces adsorbed H$_2$. Various experimental techniques are employed to collect evidence for hydrogen adsorption, isotope selectivity and isotope exchange mechanism in Cu(I)-MFU-4l. Adsorption isotherms and thermal desorption spectroscopy (TDS) are used to independently determine adsorption enthalpies, and confirm that the system follows the Langmuir model, which is employed in atomistic calculations. Inelastic neutron scattering (INS) confirms the MOF structure and provides the atomistic structure of adsorbed H$_2$. TDS and INS proof the isotope exchange mechanism. A consistent picture is obtained when combining the experiments with first principles calculations. While we have restricted our experiments to available and harmless H$_2$ and D$_2$ gases, predictions for T$_2$ and mixed isotopologues are presented on grounds of first-principles calculations, which are validated against the experiments on H$_2$ and D$_2$.

## Results

**Structure revalidation by INS.** First, we reconfirmed the structure of our Cu(I)-MFU-4l samples by INS measurements, which match the calculated Γ point phonon density-of-states[14] (Supplementary Fig. 1).

**Thermal desorption spectroscopy.** The selective adsorption after exposure to H$_2$/D$_2$ isotope mixtures was directly measured utilizing TDS: after exposure at a fixed temperature (exposure temperature, $T_{ex}$) the free gas is evacuated and the sample is cooled down to 20 K in order to preserve the adsorbed state. Fig. 2a,b show TDS spectra in the desorption temperature ($T_{des}$) range of 15–210 K after exposure to a 1:1 of H$_2$/D$_2$ isotope mixture at different $T_{ex}$ (20, 30, 50 and 100 K). The spectra can be divided into a low-temperature regime with two maxima at 30 and 50 K, and, well-separated, a high-temperature regime with another maximum for H$_2$ at 165 K and for D$_2$ at 180 K (Fig. 2a). Figure 2b magnifies the desorption rate in the range between 100 and 210 K and shows the effect of increased $T_{ex}$: while H$_2$ dominates at $T_{ex}$ = 20 K, at $T_{ex}$ = 50 K the amounts of H$_2$ and D$_2$ are about the same, whereas at $T_{ex}$ = 100 K the TDS spectrum is governed by D$_2$.

Then, in a stepwise gas loading, the sample is exposed first to H$_2$ (5 mbar for 10 min) and afterwards to D$_2$ (5 mbar, 10 min, yielding an overall isotopologue mixture with a ratio of 1:2.5 of the exposed gas). If this is performed at $T_{ex}$ = 50 K, H$_2$ dominates the high temperature TDS spectrum, while conversely D$_2$ governs for $T_{ex}$ = 100 K (Fig. 2c). If the loading sequence is reversed, D$_2$ dominates for $T_{ex}$ = 50 K, and remains the dominant adsorbate at $T_{ex}$ = 100 K, even though in this case the sample is exposed to 2.5 times more H$_2$ than D$_2$ (Fig. 2d and methods TDS).

**Isosteric heat of adsorption.** The isosteric heat of adsorption $\Delta Q$ for H$_2$ and D$_2$ was measured employing two independent techniques, that is, gas adsorption isotherms and TDS. Adsorption isotherms of H$_2$ and D$_2$ are determined (Supplementary Fig. 2) at $T$ = 173 K and above, where only strong adsorption sites are occupied in the low pressure regime. The isosteric heat of adsorption is determined directly from a van't Hoff plot (Supplementary Figs 3 and 4), resulting in $\Delta Q = 35.0 \pm 0.3$ ($32.7 \pm 0.3$) kJ mol$^{-1}$ for D$_2$ (H$_2$) (all error bars refer solely to the mathematical analysis, for details of the calculations see Supplementary Note 1). Using the same experimental data, the adsorption enthalpy $\Delta H = -\Delta Q$ can be calculated analytically using the Langmuir model (Supplementary Fig. 5, Supplementary Tables 1 and 2), yielding almost equivalent results as for the (more general) van't Hoff plot: $\Delta H = -35.0 \pm 0.1$ ($-32.6 \pm 0.2$) kJ mol$^{-1}$ for D$_2$ (H$_2$). We have also determined the desorption energy directly from the TDS of a 1:1 mixture of D$_2$ and H$_2$ at exposure temperature of 100 K (Supplementary Note 2, Supplementary Fig. 6, Supplementary Table 3). Due to the weak H$_2$ signal only the value for D$_2$ ($33.2 \pm 0.52$ kJ mol$^{-1}$) is quantitatively correct, and significantly higher than the value for H$_2$ ($24.8 \pm 0.90$ kJ mol$^{-1}$). Finally, DFT-D3 calculations give $\Delta Q = -33.7$ ($-30.5$) kJ mol$^{-1}$ for D$_2$ (H$_2$) (see Supplementary Table 4). All methods are compared in Supplementary Table 5.

**Inelastic neutron spectroscopy.** Figure 3a compares the INS spectra for low and high (about three times larger) H$_2$ gas loadings. The loading was performed at 77 K, and the temperature was decreased slowly over about 1 h to the base temperature of 5 K, at which both spectra were recorded. A blank, corresponding to the total signal coming from the aluminum sample holder and the MOF, has been subtracted in all shown spectra. The INS for low loading (black) exhibits a strong double peak at

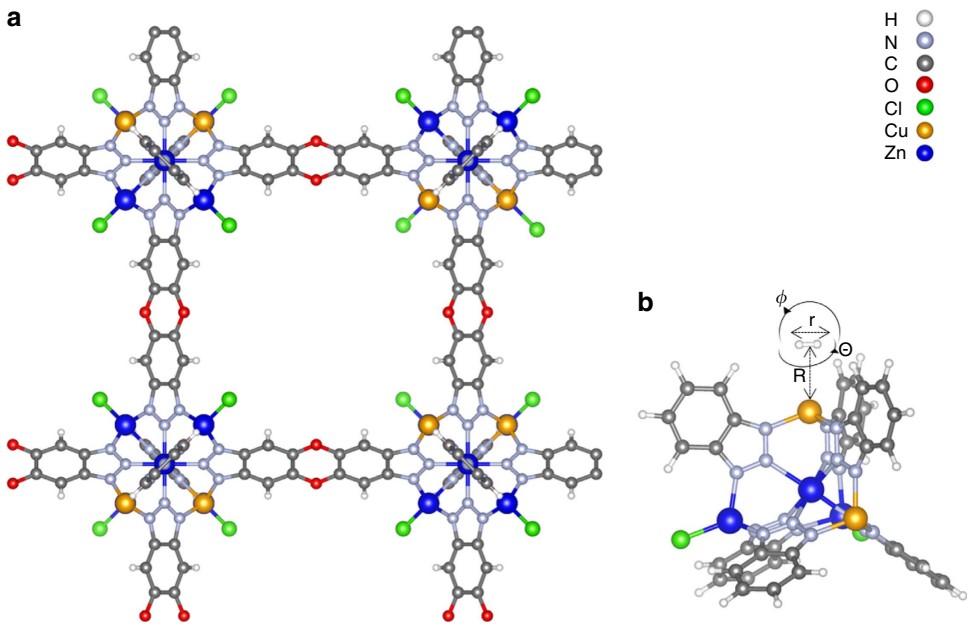

**Figure 1 | Structural model of Cu(I)-MFU-4l.** $H_2$ adsorption in Cu(I)-MFU-4l: cluster showing a full pore (**a**); computational model (**b**). $R$ is the distance between Cu and $H_2$ centre; $r$ is the H–H distance of adsorbed $H_2$; $\phi$ is the angle of rotation of adsorbed $H_2$ in the plane normal to $R$, and $\Theta$ is the angle of out-of-plane rotation. Colour scheme: white—H, light blue—N, grey—C, red—O, green—Cl, orange—Cu, dark blue—Zn.

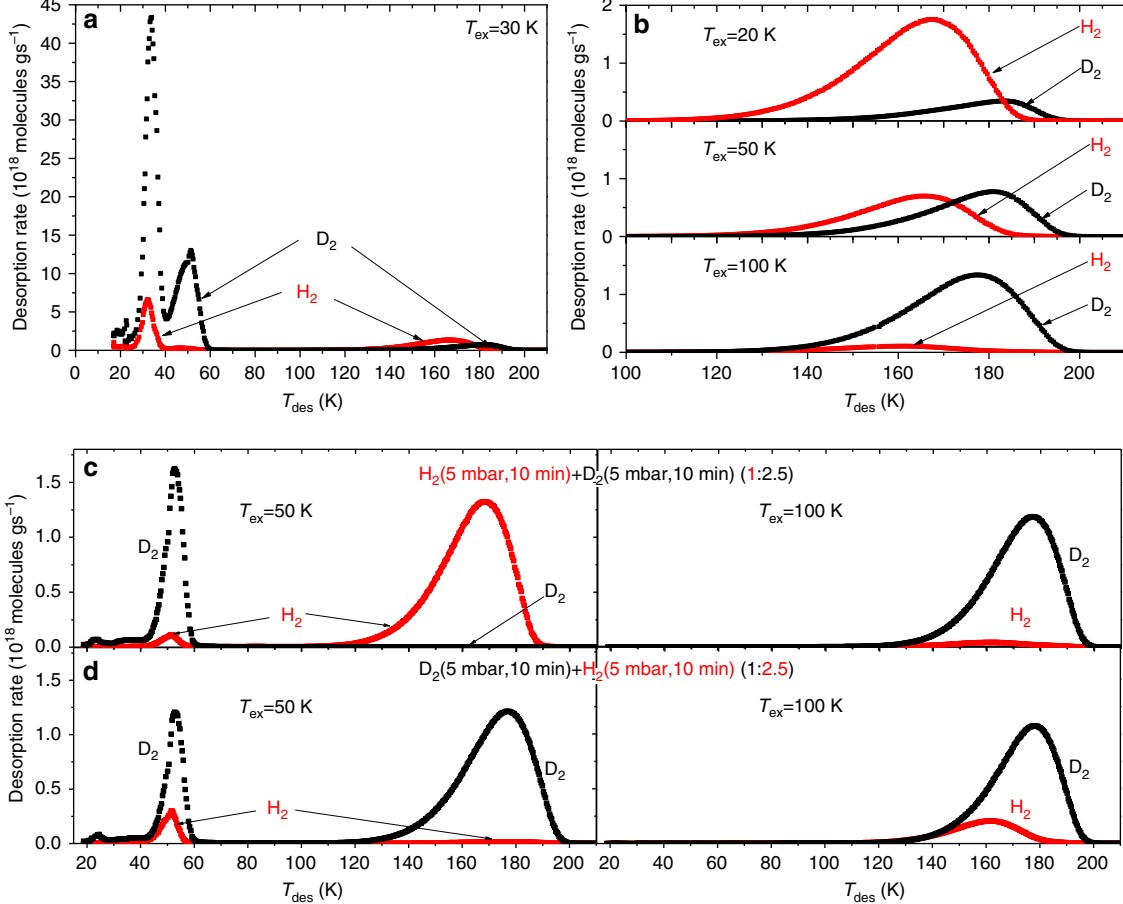

**Figure 2 | Thermal desorption spectra of adsorbed $H_2$ and $D_2$.** TDS spectra after exposure to a 10 mbar 1:1 $H_2$ (red)/$D_2$ (black) mixture for 10 min at different exposure temperatures $T_{ex}$, showing the whole (**a**) and high-temperature (**b**) range. Stepwise loading at $T_{ex} = 50$ and 100 K: first $H_2$ (5 mbar, 10 min), then 5 mbar of $D_2$ was added for another 10 min (**c**) and reverse loading sequence (**d**).

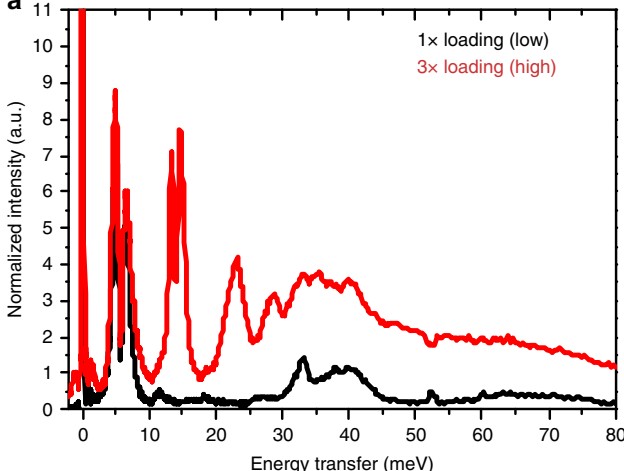

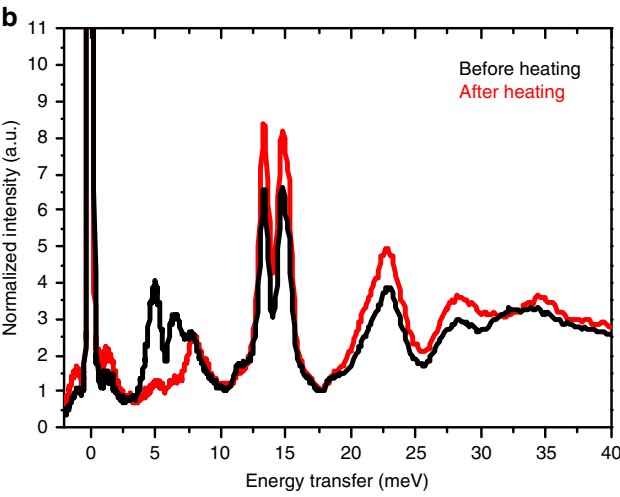

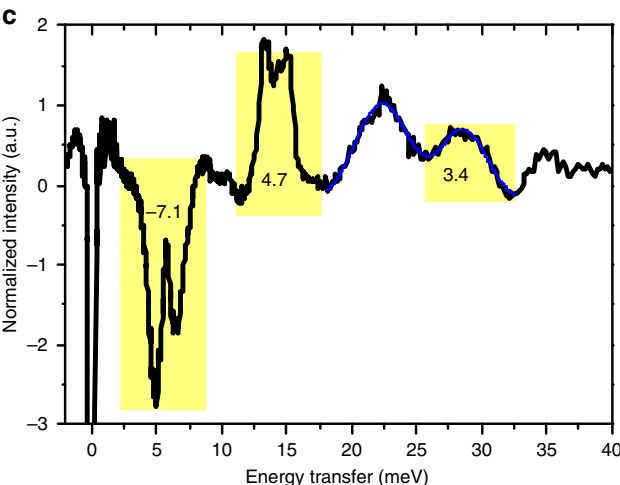

**Figure 3 | INS of Cu(I)-MFU-4l dosed with H₂ and D₂.** (a) In the black spectrum the quantity of gas dosed was not enough to saturate the first adsorption sites. The second dosing (red spectrum) saturates the first site and populates the second sites (the double peak 13.4 and 14.7 meV). (b) INS spectra with D₂ gas dosed in addition to H₂ at 70 K, before (black) and after (red) heating to 200 K, see Supplementary Fig. 7 for more spectra measured during heating. (c) Subtraction of the spectra recorded at 5 K before and after the exchange of H₂ through D₂. The exchanged amount is shown by the negative sign, while H₂ moved to the weaker adsorption sites as indicated by the intensity gain.

4.9 and 6.5 meV, while for higher loading (red) additional signals appear at 13.4, 14.7 and 23.0 meV.

In the experiment resulting in Fig. 3b, about half of the $H_2$ gas was pumped out at 100 K and approximately the same amount of $D_2$ gas was added at 70 K. Immediately after loading, the sample was cooled to 5 K and a neutron spectrum was measured (Fig. 3b and Supplementary Fig. 7, black). The two maxima of the strong adsorption site have strongly decreased, whereas the peaks reflecting the weak adsorption sites have almost the same intensity. Note that since $D_2$ has a much lower cross-section compared to $H_2$, adsorption sites occupied by $D_2$ will not be visible in the spectra. In addition, spectra of the gas mixture have been recorded at different temperatures (40, 77, 110, 150 and 200 K). For temperatures 5–110 K the spectra are plotted in Supplementary Fig. 7. After the 200 K measurement the sample was cooled again to 5 K and the spectrum was re-measured. The two maxima at the strong adsorption site vanished completely, indicating that during heating to 200 K all $H_2$ on this site were exchanged by $D_2$. At the same time, the intensity is increased at the weak adsorption site (Fig. 3b). Figure 3c shows the difference between the 5 K measurement before and after the heating procedure. Negative values indicate a decrease of occupation by $H_2$, that is, an exchange by $D_2$. The decrease on the strong adsorption site is approximately the same as the increase on the weak adsorption sites.

**Raman spectroscopy**. Raman measurements (Supplementary Note 3, Supplementary Fig. 8) show free $H_2$ (and $D_2$), but no strongly shifted signals of adsorbed species can be observed in the sample. At low temperatures, a signal, red-shifted by 0.2–0.7% with respect to the Q(0)-line of the free gas, appears.

**First principles and model calculations**. The Born-Oppenheimer first principles calculations (details are given in Methods and thorough method validation data are given in Supplementary Note 4, Supplementary Figs 9–11 and Supplementary Tables 6–11) show a potential energy surface with pronounced anharmonicity, both for the bonding of the $H_2$ centre to the Cu(I) site (associated bond length denoted as $R$) as well as for the intramolecular covalent bond of the adsorbed $H_2$ species (associated bond length denoted as $r$, see Fig. 1). The vibration of the isotopologue centre along bond parameter $R$ is associated with frequency $v_R$, while $v_r$ is related to intramolecular vibration along $r$ (Fig. 1). We assume that the associated potential energy curves are essentially decoupled and fit each of them to a Morse potential. This allows capturing quantum-mechanically the isotope effect by analytical solution of the Schrödinger equation for the two vibrations associated with the adsorbed hydrogen isotopologues, resulting in frequencies $v_R$ and $v_r$, associated zero point energies (ZPE) $E_0 = \frac{1}{2}hv - \frac{(\frac{1}{2}hv)^2}{4D}$ and in the probability densities describing both vibrational modes (Table 1, Fig. 4).

Rotational levels associated with the angle $\varphi$ (see Fig. 1) are estimated using the effective potential of a perturbed planar rotor $V(\varphi) = 2q \cos(2\varphi)$. The resulting Schrödinger equation is the Mathieu differential equation for which the solution is known. Contrary to the free two-dimensional rotor, the rotational levels split, and the value for $q$ is directly obtained from the INS data of the first rotational excitation: $2q = (6.5–4.9)$ meV; $q = 0.8$ meV. First principles calculations suggest a larger value of $q = 5.4$ meV; however, this still rather small value is well below the intrinsic accuracy of the method/model combination. The position of the INS signal gives the bond length expectation value of the adsorbed $H_2$ molecule $\langle r \rangle = 0.85$ Å (first principles calculations: $\langle r \rangle = 0.83$ Å).

**Table 1 | Expectation values and frequencies**

| Isotopologue | $\langle r \rangle$, Å | $\langle R \rangle$, Å | $\nu_r$, cm$^{-1}$ | $\nu_R$, cm$^{-1}$ |
|---|---|---|---|---|
| H$_2$ | 0.830 | 1.738 | 3,320 | 1,363 |
| D$_2$ | 0.824 | 1.712 | 2,380 | 1,077 |
| T$_2$ | 0.821 | 1.701 | 1,956 | 920 |
| HD | 0.827 | 1.726 | 2,893 | 1,244 |
| DT | 0.822 | 1.706 | 2,734 | 1,005 |
| HT | 0.826 | 1.721 | 2,179 | 1,195 |

Expectation values $\langle r \rangle$ of bond length X1–X2 of adsorbed hydrogen isotopologues (X = H, D, T), expectation value $\langle R \rangle$ of the distance between Cu and the centre of the adsorbed X1X2 isotopologues, ($\nu_r$) stretching frequency of adsorbed isotopologues, ($\nu_R$) up-down translational frequency of adsorbed isotopologues, all calculated from the Morse fit of the potential energy surface. For definition of geometrical parameters, see Fig. 1.

Due to the high heat of adsorption of H$_2$ at the undercoordinated Cu sites (Supplementary Table 5), a remarkably high temperature of 200 K is needed to release all gas molecules from the samples. The TDS shows a significant isotope effect with an upshift of the high-temperature desorption maximum of ~15 K for D$_2$. The adsorption enthalpy difference between H$_2$ and D$_2$ is governed by the zero-point energy, which in turn is related to the vibrational frequency of the adsorbed molecules. As both isotopologues are subject to the same spring constant $k$ and differ by a factor of 2 in mass, within the harmonic approximation a rough estimate for $\nu_R$ can be obtained ($E_0 = \frac{1}{2} h\nu = \frac{1}{2} \hbar \sqrt{\frac{k}{m}}$). The experimentally observed 2.5 kJ mol$^{-1}$ difference in zero-point energy between H$_2$ and D$_2$ is then equivalent with $\nu_R = 1,427$ cm$^{-1}$ for H$_2$, compared to $\nu_R = 1,363$ cm$^{-1}$ obtained directly from the first principles calculations (including anharmonicity). Thus, the calculated probability density describing the position of the H$_2$ centre as function of the distance to the Cu(I) site is consistent with the difference in H$_2$ and D$_2$ adsorption enthalpy obtained from our experiments. This has a direct implication on the intramolecular vibration ($\nu_r$): in the interval of significant probability density $|\Psi(R)|^2$ the calculated $\nu_r$ value varies by several 100 cm$^{-1}$ (Fig. 4b). Thus, a strongly shifted $\nu_r$ signal, characterizing the chemically activated hydrogen molecule, would be so broad that it cannot be detected in the Raman experiment.

It is, however, possible to determine the activated H–H bond length at the strong Cu(I) site directly from the double peak (4.9 and 6.5 meV) in the INS spectrum. Application of the model potential of an isolated two-dimensional rotor to interpret the INS data gives a hydrogen bond length of about 0.85 Å, slightly larger than the 0.83 Å obtained from the first principles potential energy surface scan. At the same time, the potential energy surface shows a double peak separated by 1.6 meV, which coincides with the potential barrier of the H$_2$ rotation, with H$_2$ oriented normal to vector R. The calculated barrier is significantly higher (5 meV), which is, however, below the intrinsic accuracy limit of the DFT calculation. However, our calculations are capturing the salient interactions and physics, providing the requisite insights.

For higher loadings an additional double peak appears at 13.5 and 14.7 meV, which can be assigned to the rotational transitions of weakly bound H$_2$ from the ground state $J = 0$ to the first excited rotational state $J = 1$ (ref. 15), with lifted degeneracy between ($J = 1$, $M = 0$) and ($J = 1$, $M = +- 1$) with an effective rotational constant of B = 7.0 meV. The rotational constant of free H$_2$ is B = 7.35 meV, indicating that only a small perturbation occurs at the weak adsorption sites. The other signals of the INS spectrum give further insight into the vibrational modes of the adsorption complex, also in dependence of the nuclear spin polarization (see Supplementary Fig. 7).

A very important result of this work is the strong isotope exchange mechanism occurring at temperatures of 90 K and higher. For lower temperatures, the adsorption process is kinetically controlled: the TDS data show that light-weight H$_2$ with its higher diffusion constant enters the material first and dominates the TDS, where most of the Cu(I) adsorption sites are occupied by H$_2$ at a first-come first-serve basis, and where a barrier prevents molecular exchange at the strongly attractive sites. At about 90 K, though, H$_2$ is replaced by slower, but more strongly adsorbing D$_2$, reflecting a thermodynamically controlled mechanism. Thus, Cu(I)-MFU-4$l$ allows the selective capture of heavier isotopologues from the gas phase. This exchange mechanism is observed both in the TDS and in the INS measurements and occurs at temperatures higher than 90 K until all gas is desorbed from the samples. Finally, no isotope exchange

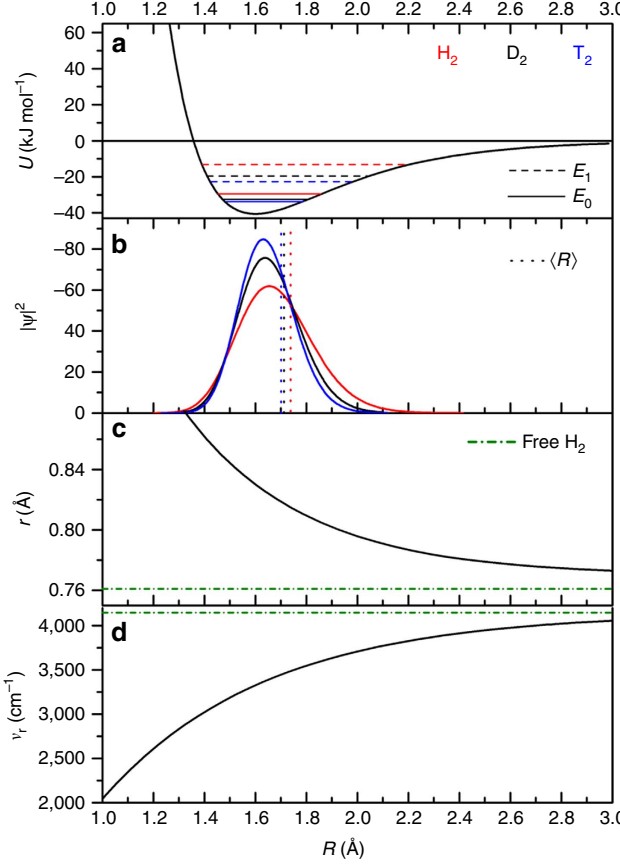

**Figure 4 | Properties of adsorbed molecular hydrogen at the Cu(I) sites.** (**a**) The interaction potential $U$ of H$_2$ as function of distance $R$ of the molecular centre of H$_2$ from the Cu(I) site. The curve fits to a Morse potential, and the vibrational energy levels $E_0$ (zero point energy) and $E_1$ (first excitation) are given for the adsorbed hydrogen isotopologues H$_2$ (red), D$_2$ (black) and T$_2$ (blue). (**b**) Probability density $|\Psi(R)|^2$ of the adsorbed hydrogen isotopologues (same colour code). Probability values are given as dotted vertical lines. (**c**) Intramolecular H$_2$ distance $r$ as function of $R$. (**d**) Frequency of the vibrational intramolecular stretching mode $\nu_r$ of the adsorbed isotopologues. In **c**,**d**, the value for free H$_2$ (green dash-dotted line) are given for comparison.

## Discussion

The combination of the various experimental results and first principles calculations allows a consistent interpretation, with the conclusion that Cu(I)-MFU-4$l$ is indeed a material that allows the capture of heavy isotopologues from a hydrogen gas mixture.

**Table 2 | Predicted selectivity of hydrogen isotopologues (thermodynamic equilibrium) at the Cu(I) sites as a function of temperature.**

| Temperature (K) | $S(D_2/H_2)$ | $S(HD/H_2)$ | $S(DT/D_2)$ | $S(T_2/D_2)$ |
|---|---|---|---|---|
| 80 | 36.9 | 5.8 | 2.3 | 5.6 |
| 90 | 21.7 (7.1) | 4.5 | 2.1 | 4.5 |
| 100 | 14.2 (11.1) | 3.7 | 1.9 | 3.7 |
| 120 | 7.6 | 2.8 | 1.7 | 2.9 |
| 130 | 6.1 | 2.5 | 1.6 | 2.6 |
| 140 | 5.0 | 2.3 | 1.53 | 2.4 |
| 150 | 4.2 | 2.1 | 1.48 | 2.2 |
| 160 | 3.6 | 2.0 | 1.44 | 2.1 |
| 180 | 2.8 | 1.8 | 1.37 | 1.9 |
| 200 | 2.4 | 1.6 | 1.33 | 1.8 |
| 220 | 2.0 | 1.53 | 1.29 | 1.7 |
| 240 | 1.8 | 1.45 | 1.26 | 1.6 |
| 260 | 1.6 | 1.39 | 1.24 | 1.5 |

The selectivities are given for 1:1 mixtures. Experimental values are given in parenthesis. Note that experimental values are not necessarily at thermodynamic equilibrium.

to form mixed isotopologue HD is observed, even though INS and first principle calculations confirm highly activated hydrogen at the Cu(I) sites. However, as intramolecular bond breaking energy is about 13 times higher (DFT-D3) compared to the adsorption enthalpy of $H_2$ in Cu(I)-MFU4l, such exchange should not be expected at the time scale of our experiments.

Table 2 shows the selectivity at the strong adsorption site, determined from the TDS measurements, together with predicted selectivities of the most important hydrogen isotopologues in this material based on first principles calculations and backed up with the experimental data obtained for $H_2$ and $D_2$. The $D_2$-over-$H_2$ selectivities reported here are the highest known to date, higher than the S = 12 at 60 K for CPO-27(Co)[7], and significantly higher than reported values on carbon nanopores, which do not exceed a selectivity of 3 at 77 K[16]. They are not directly comparable to recently reported H-over-D selectivities of 10 at room temperature observed for sieving at two-dimensional crystals[6]. It is evident that the material is promising for selection and purification of HD and $D_2$ from an isotope mixture, but also for capturing $T_2$ and DT.

In conclusion, we have shown that Cu(I)-MFU-4l is a material suitable for the capture of heavy hydrogen isotopes from the gas phase. The material shows a remarkably strong hydrogen adsorption enthalpy with pronounced isotope effects, even at temperatures well above 100 K. Moreover, in contrast to previously studied MOFs such as CPO-27(Co)[7], it further shows a strong isotope exchange effect that allows the enrichment of heavy isotopologues from low-concentration phases. Thus, with its high selectivity, it is a very promising material to produce high-purity deuterium, and for capturing tritium from low-concentration hydrogen gas mixtures. The latter could significantly enhance our ability to treat radioactive waste, and, at the same time, produce precious $^3He$ and tritium fusion reactor fuel. For large-scale application the density of these strong adsorption sites should be increased to allow for higher loadings at high temperature.

## Methods

**Thermal desorption spectroscopy.** TDS is a very sensitive technique to characterize gas–solid interactions. Only a small amount of sample (here 2.2 mg) is necessary for our especially designed TDS apparatus[17]. Before measurements the sample was outgassed at 453 K under high vacuum to remove solvent molecules from the Cu(I) sites. For each measurement the sample was cooled to a certain exposure temperature. The gas was dosed to the sample by two different methods:

(1) an equimolar mixture of the two isotopes was prepared outside the sample chamber. Then the sample was exposed to this mixture of 10 mbar for 10 min, (2) 5 mbar of one isotope was introduced to the volume of the pre-chamber and sample chamber for an exposure time of 10 min. The sample chamber was closed and the remaining gas in the pre-chamber has been evacuated. Sample chamber and pre-chamber were connected again and 5 mbar of the other isotope was added for an exposure time of 10 min. This stepwise loading procedure results in a final mixture of 1:2.5 (1st:2nd gas). The following steps of the measurement procedure are independent of the gas loading method. After the exposure the not adsorbed gas was pumped out at the exposure temperature and the sample was cooled down to 20 K. A constant heating rate of 0.1 K s$^{-1}$ induces desorption of the gas, which is then detected by a mass spectrometer. A typical TDS spectrum consists of several maxima where the temperature of the maxima indicates the strength of the adsorption site and the area under the peaks is proportional to the adsorbed gas amount. Furthermore, the output of the mass spectrometer is calibrated for $H_2$ and $D_2$ by measuring an alloy ($Pd_{95}Ce_5$) with known $H_2$ ($D_2$) content[18].

**Hydrogen adsorption enthalpy and entropy.** Cu(I)-MFU-4l was prepared as described previously[13] by heating Cu(II)-MFU-4l-formate for 1 h at 180 °C under high vacuum. Adsorption isotherms were measured with a BELSORP-max instrument combined with a BELCryo system. Between the adsorption measurements the sample was degassed for 2 h at 150 °C under high vacuum ($P < 10^{-4}$ mbar). Adsorbed amounts are given in cm$^3$ g$^{-1}$[STP], where STP = 101.3 kPa and 273.15 K. The estimation of adsorption enthalpies is described in the SI.

**Inelastic neutron scattering.** The INS spectra were recorded at Oak Ridge National Laboratory at the Spallation Neutron Source on the indirect geometry spectrometer Vision (beamline 16B). The activated Cu(I)-MFU-4l sample (1.044 g) was loaded in a helium-filled glove box into an aluminum cell and connected to a gas handling system. The sample was evacuated with a turbo-molecular pump at ambient temperature overnight. Measurements of the evacuated sample and sample with gas inside are performed at different temperatures (5–200 K) controlled through heaters attached to the sample cell and on the Closed Cycle Refrigerator. $H_2$ was converted to para $H_2$ at 20 K by using $Fe_3O_4$ as catalyst and loaded volumetrically to the sample at about 70 K.

**Theory.** London dispersion corrected density-functional theory (DFT-D3) calculations are carried out on a relatively large cluster model of the Cu(I) adsorption site, reflecting the Langmuir model that is valid in the low-pressure regime (see Model I-Cl in Supplementary Fig. 9). The DFT-D3 approach (hybrid functional PBE0-D3, cc-pVTZ-PP basis set for Cu, def2-TZVPP for the interacting $H_2$ and def2-TZVP for all other atoms) has been validated against coupled cluster theory including single, double and partially triple excitations (CCSD(T)) calculations (Supplementary Table 6), and has also been used elsewhere[7,19]. Two open Cu(I) metal centres per formula unit are statistically distributed in Cu(I)-MFU-4l[13]. In order to reduce the computational effort we used only one Cu(I) metal site per secondary building unit (SBU), as test calculations confirmed that these sites act as individual attraction sites. Influence of formate anions (bound to the Zn atoms) and utilization of a bigger cluster model had negligible effect on the adsorption properties of the Cu(I) site (see Supplementary Tables 7 and 8). However, inclusion of long-range interactions using a high-/low-level subtraction scheme (cluster as described above, periodic plane wave calculation at the PBE-D3 level of theory (Supplementary Note 5, Supplementary Table 9) lowers the adsorption enthalpy by 0.7 kJ mol$^{-1}$. An energy decomposition analysis shows that 62% of the interaction energy is due to Coulomb interaction (Supplementary Note 6, Supplementary Table 10, Supplementary Fig. 11).

Excellent agreement with experiment (within 1 kJ mol$^{-1}$ for adsorption energies) is achieved if the strongly anharmonic potential energy surface describing the guest–host interaction is taken into consideration. This allows the accurate prediction of structural and thermodynamic data for all six isotopologues involving H, D and T.

The selectivities of different hydrogen isotopologues are calculated following the approach suggested by Sillar et al.,[20] which was adopted for strong adsorption centres earlier[7]. Following this approach, the three translational vibrations and the stretching mode were converted to four vibrations relative to the adsorption site. Noteworthy, the rotation of adsorbed $H_2$ along angle $\theta$ (see Fig. 1, Supplementary Fig. 11) must be treated as vibration (1,297 cm$^{-1}$), as the rotational barrier is 159.4 kJ mol$^{-1}$. This also motivates application of the Mathieu equation for the rotation around $\varphi$, which allows the interpretation of the INS data. Calculated adsorption enthalpies (and its individual contributions) are given in Supplementary Table 11.

**Data availability.** The data that support the findings of this study are available from the corresponding author upon reasonable request.

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

## Acknowledgements

Financial support by Deutsche Forschungsgemeinschaft (SPP 1362) and the European Commission (ERC-StG 256962) is gratefully acknowledged.

## Author contributions

All authors contributed extensively to the presented work. D.D. and D.V. provided the materials, measured and analysed the adsorption isotherms, I.W. performed the TDS and Raman experiments with support from S.M.S., H.-H.K. and M.L.T. I.S. and A.M. performed the first principles and model calculations with support from T.H. Y.C. and A.J.R.-C. performed the INS measurements and analysed it together with L.L.D., I.W., M.H. and T.H. TDS and Raman data have been analysed by I.W., M.H., G.S. and T.H. T.H., I.W. and M.H. wrote the manuscript in close contact with all authors.

## Additional information

**Competing financial interests:** The authors declare no competing financial interests.

**Publisher's note**: 

