## [Peer Review File · Nature Communications]

Reviewers' Comments:

Reviewer #1 (Remarks to the Author)

Authors report capture of D2 and T2 using metal organic frameworks. The reviewer felt the paper is too preliminary with limited experimental data to publish in high impact journal like Nature Communications. Therefore i do not recommend for publication in Nature Communications, might be suitable for Scientific reports with minor changes.

1) Reviewer recommend the use of D2 and T3 through out the text, It is difficult to read the text where it starts and ends with "T" Either indicate T isotope or T3 isotope, the same applies to D2 isotope

2) Authors need to provide some spectroscopic evidence such as Infra red spectroscopy. Several authors published papers on the use of IR to distinguish between these two isotopes. See ref 8 or Prof. Chabal papers.

3) It would be helpful for readers to update the Table with sensitivities of D2/H2 of MOF vs Graphene related materials.

Reviewer #2 (Remarks to the Author)

The authors use a diverse, effective set of techniques to characterize the separation of hydrogen isotopes (H, D) in the considered MOF platform. They use their insights and analysis to predict its utility for T, tritium. The proposed process has the advantage of selectively capturing the rarer heavier isotopes from a hydrogen mixture. Sorting out the mechanisms through the combination of experimental techniques and theory is quite attractive.

I think this paper is a reasonable candidate to be published. The manuscript needs to be presented in a more compelling form with the case for the materials importance better presented early on. In the introduction the authors need to explain why this material is distinct / superior to the similar performance in the other MOFs with similar properties like CPO-27(Co) and M-MOF-74 metal variants. They appropriately reference the other MOFs but need to explain why the new material with similar performance is of sufficient interest for a high profile journal. They mention it is the isotope enrichment capacity that stands out in the conclusion but this needs to be elaborated on and presented in the introduction as an important feature and distinct from extant material / MOF performance.

The authors need to make clearer physical arguments to the readers that provide insight and clarity into the important physical measurements they have performed. Basically, I am suggesting that the paper needs polishing and enhancement but the science is sound. It is an effective use of a variety of measurement techniques combined usefully with theoretical insights. The material is predicted to also have useful properties for T separations that are inferred from the data and theory that has technological utility. (Another minor point is the authors should address whether the MOF has any practical limitations for the imagined uses.)

The use of the calculated vibrational density of states compared to neutron scattering is a useful characterization method. Perhaps the authors can comment on and reference other such comparisons to put the level of the observed agreement in context. The spectra looks reasonable but is this necessary and / or sufficient?

The theoretical methods used provide the requisite insight and have sufficient accuracy.

I recommend the authors say something like "The theoretical calculations are capturing the salient interactions and physics, providing the requisite insights."

instead of, "However, overall, experiment and theory are in excellent agreement." The later sentence is not informative and is neither a necessary or sufficient condition for the theory to be usefully relevant.

Both of the below sentences need to be clarified. I do not understand the first and while the second is clearer I am not sure why no experimental signal is possible:

Thus, the probability density describing the position of the H₂ centre as function of the distance to the Cu site is highly justified. The Raman shift of the intramolecular frequency of H₂ varies strongly (several 100cm⁻¹, see Figure 4b) as function of R. As the adsorbed H₂ molecule is strongly delocalized (cf. the probability density in Figure 4b) a strongly shifted ν_R signal indicating the activated hydrogen molecule would be so broad that it cannot be detected in experiment"

I also think a brief, simple explanation of how the zero point energies change the isotopic binding should be included - this paper is for a broader audience. Such a simple explanation would serve to inform the interesting nature of the results and focus readers on this simple quantum phenomenon (physical scientists may even forget the Born-Oppenheimer surfaces are not mass dependent).

The colors not closely matching the legend in Figure 1 needs to be addressed. It is a critical figure and it makes it difficult to interpret.

In the supplemental information the authors should also show the data in Figure S2 overlapped in a separate panel. The authors should also approximate the isosteric heat of sorption directly from the temperature dependent sorption data using Clausius-Clapeyron analysis. This is in a sense less model dependent and serves as a consistency check.

In Figure S4 the caption line "The modified vibration occurs due to the reduced degrees of freedom of the physisorbed H₂ and D₂." is unclear. Do they mean the vibrational frequency shift occurs due to the attractive sorbate /sorbent interaction?

There are other language issues, even the first sentence:

"The metal-organic framework Cu(I)-MFU-4l captures heavy hydrogen isotopes deuterium and tritium from hydrogen gas phase mixture with D₂/H₂ separation factor of 11 at 100 K." isn't standard English. The text needs to be cleaned up for a broad audience in a high profile journal.

In this sentence: "The Born-Oppenheimer first principles calculations show a strongly anisotropic potential energy surface, both for the distance of the H₂ centre to the Cu(I) site (denoted as R) as well as for the bond length of the adsorbed H₂ species (denoted as r, see Figure 1)." The language here is not very clear although I think the intent is to refer to mapping the potential energy surface for the two coordinates defined in Figure 1 and they find an anisotropic surface of some sort. It is unclear what anisotropic means for a one dimensional coordinate. Clarity is required here. Is it exceptional anharmonicity? If so compared to what?

I am not sure what this means: "Thus, the probability density describing the position of the H₂ centre as function of the distance to the Cu site is highly justified."

The lack of clear, precise language throughout detracts from the important science.

In sum, the manuscript has many nice features. For it to be appropriate for a wide audience a clear compelling case needs to be made that this particular MOF has important distinct features from those previously characterized. None of the work is exceptional on its own - the effect is known in similar materials and the characterization techniques are used routinely. It is the combination of insightful experiments and theory that make the paper interesting. For it to be publishable a more polished manuscript with crisp, clear arguments and insights needs to be presented. Note, in my opinion, I have little issue with the data and the presentation of the results is adequate if not exceptional.

Reviewer #3 (Remarks to the Author)

This paper reports the captures of heavy hydrogen isotopes on MOF Cu(I)-MFU-4l. The experimental evidence is restricted to H₂ and D₂. The H₂/D₂ separation factor was 11 at 100K.

1) The hydrogen adsorption enthalpy is reported as 32 kJ.mol⁻¹. This value is very high, if not the highest value obtained so far and the standard deviation for the value should also be reported. As confirmation of this very high value, the authors should report an independent experimental measurement of the adsorption enthalpy for D₂, which should be slightly higher or the same as H₂, within experimental uncertainty. It is very important that such a high value for adsorption enthalpy is supported by additional experimental data.

2) Applications of first principles calculations have been used to predict performance for the capture of tritium. However, the paper does not give experimental evidence to support the predictions. Therefore, I recommend a change in title for the paper to make it clear that the experimental results are restricted to H₂ and D₂.

3) A significant amount of literature is available on isotope effects. Thermal desorption spectra as shown in Fig 2 have been reported previously by some of the authors of this paper - see Ref 16 etc. Therefore the most interesting aspect of this paper is the very high enthalpy obtained for MOF Cu(I)-MFU-4l and the explanation for potential selectivity.

4) Page 8- last 4 lines. Comparison between INS and first principles calculations suggest a slight underestimation of R in the latter, which is consistent with a larger barrier. This can be due to the relatively small cluster model, which neglects long-range crystal interactions.

Please quantify the errors and the likely limitations for model predictions for T₂ to increase confidence in the accuracy of the calculations since all models have approximations.

5)Page 9 line 24

Interestingly, no isotope exchange to form mixed isotopologue HD has been observed, even though INS and first principle calculations confirm highly activated hydrogen at the Cu sites.

This appears to be a contradiction for the model, which requires further explanation.

Response to Reviewers:

Reviewers' comments:

Reviewer #1 (Remarks to the Author):

Authors report capture of D2 and T2 using metal organic frameworks. The reviewer felt the paper is too preliminary with limited experimental data to publish in high impact journal like Nature Communications. Therefore i do not recommend for publication in Nature Communications, might be suitable for Scientific reports with minor changes.

We thank the reviewer for reading the manuscript and would like to understand what exactly is preliminary in this work. The scientific point is, in our opinion, well-substantiated and scientifically justified. We hope that the clarifications as made during this review process convince Reviewer #1 of this work.

1) Reviewer recommend the use of D2 and T3 through out the text, It is difficult to read the text where is starts and ends with "T" Either indicate T isotope or T3 isotope, the same applies to D2 isotope

We do not understand this comment, possibly it is due to a loss of formatting of super- and subscripts. According to IUPAC hydrogen isotopes are to be labelled as protium: ^1H , deuterium: ^2H , tritium: ^3H . However, this nomenclature is too clumsy for addressing isotopologues of hydrogen molecule H_2 , or $^1\text{H}_2$. Therefore, we use the nomenclature that is established in the literature: $^1\text{H}=\text{H}$, $^2\text{H}=\text{D}$, $^3\text{H}=\text{T}$, leading to hydrogen molecule isotopologues H_2 , HD, HT, D_2 , DT and T_2 . We explicitly introduced this nomenclature in the first sentence of the main text in the manuscript.

2) *Authors need to provide some spectroscopic evidence such as Infra red spectroscopy. Several authors published papers on the use of IR to distinguish between these two isotopes. See ref 8 or Prof. Chabal papers.*

IR spectroscopy is indeed a very useful technique for determining adsorbed hydrogen in samples such as MOFs. There are two vibrational modes that are characteristic for hydrogen isotopologues: the one related to the molecular adsorption process (labelled ν_{IR} in the previous version of the manuscript in order to denote IR activity, now changed to ν_R to associate it with the mode describing the distance R of the centre of the adsorbate from the Cu site), and one that characterises the intramolecular vibration of the hydrogen isotopologue (labelled ν_R in the old version to denote Raman activity, now changed to ν_r to denote the relation to the distance r between the two hydrogen isotopes). The former one is the one that, e.g., FitzGerald and coworkers were measuring, the latter one can also be accessible in experiment. ν_r was not visible in Raman experiments, which is rationalized in detail in the manuscript, due to strong delocalization of the adsorbate and resulting line width broadening of this frequency. For measuring ν_R a special setup, i.e. a cryogenic device, is needed that none of the participating groups has available, and the success of such experiment remains to be tested due to the relatively low concentration of adsorption sites. Note that this mode is directly associated with the desorption of the molecule, thus only low-intensity measurements are possible. In short: this is a very challenging experiment, and it is not possible for us to carry out these experiments at the moment. However, as will be discussed below, this experiment is not needed to explain our results in a satisfactory manner, as explained below.

The characteristic frequency can be estimated from experiment, and this estimate is in good agreement with our DFT calculation: As outlined in the manuscript, the calculated value for ν_R is in very close agreement with a frequency estimated from experimental data, namely by the difference in adsorption enthalpies of D_2 and H_2 , which can be related to the vibration of a model system (infinite mass of the MOF, H_2 and D_2 as shapeless particles that are oscillating in a potential). We can thus estimate the frequency from the zero point energy (the employed formula is given in the revised manuscript), and this frequency agrees well with the direct calculation of the frequency modes in our DFT calculations.

3) *It would be helpful for readers to update the Table with sensitivities of D_2/H_2 of MOF vs Graphene related materials.*

We believe the referee means selectivities. We agree with the referee and updated the text, and compared the data with those of carbon-based materials. The Selectivity of graphene and other 2D crystals (which is H_2 over D_2) is also mentioned in the text.

Reviewer #2 (Remarks to the Author):

The authors use a diverse, effective set of techniques to characterize the separation of hydrogen isotopes (H, D) in the considered MOF platform. They use their insights and analysis to predict its utility for T, tritium. The proposed process has the advantage of selectively capturing the rarer heavier isotopes from a hydrogen mixture. Sorting out the mechanisms through the combination of experimental techniques and theory is quite attractive.

We thank the referee for this encouraging summary.

I think this paper is a reasonable candidate to be published. The manuscript needs to be presented in a more compelling form with the case for the materials importance better presented early on. In the introduction the authors need to explain why this material is distinct / superior to the similar performance in the other MOFs with similar properties like CPO-27(Co) and M-MOF-74 metal variants. They appropriately reference the other MOFs but need to explain why the new material with similar performance is of sufficient interest for a high profile journal. They mention it is the isotope enrichment capacity that stands out in the conclusion but this needs to be elaborated on and presented in the introduction as an important feature and distinct from extant material / MOF performance.

We thank the referee for this assessment and worked hard in improving the introduction, to highlight the outstanding performance of Cu(I)-MFU4l. We further noted that besides the isotope separation (S=11 for 100K) this materials shows a capture mechanism (that is, to replace previously adsorbed lighter isotopologue species by heavier ones) which is very promising for technological application.

The authors need to make clearer physical arguments to the readers that provide insight and clarity into the important physical measurements they have performed. Basically, I am suggesting that the paper needs polishing and enhancement but the science is sound. It is an effective use of a variety of measurement techniques combined usefully with theoretical insights. The material is predicted to also have useful properties for T separations that are inferred from the data and theory that has technological utility. (Another minor point is the authors should address whether the MOF has any practical limitations for the imagined uses.)

The requirement of Nature journals to clearly separate *Results* and *Discussion* puts quite some restrictions on presenting this work, as the results are indeed only pieces that require to be put in bigger context. The many individual measurements need to be presented under *Results*, and indeed it does not become evident in the *Results* section why these measurements have been done and how the results relate to each other. In order to address this well-justified point we made the following modifications: we have first given a little tour through the remainder of the paper at the end of the introduction,

thus preparing the reader that the individual puzzle pieces in the *Results* section will form a consistent picture that will be created in the *Discussion* section. We have then shortened the *Results* section, at the same time enriched details in the *Supporting Information*, and put more emphasis on discussion. The referee is quite right that present Cu(I)-MFU-4l is lacking a high concentration of adsorption sites, which limits technological application of this material at present. However, increasing the number of sites, or transferring the motif of the adsorption site to a different cage, is possible. We have included a corresponding statement at the end of the *Discussion* section.

The use of the calculated vibrational density of states compared to neutron scattering is a useful characterization method. Perhaps the authors can comment on and reference other such comparisons to put the level of the observed agreement in context. The spectra looks reasonable but is this necessary and / or sufficient?

This is a standard procedure that the ORNL group applies regularly to address the quality of the samples. This is highlighted in the standard reference book by P.C.H. Mitchell, S.F. Parker, A.J. Ramirez-Cuesta, J. Tomkinson which has been cited (but was not adequately explained). We have improved this passage of the text.

The theoretical methods used provide the requisite insight and have sufficient accuracy.

I recommend the authors say something like "The theoretical calculations are capturing the salient interactions and physics, providing the requisite insights."

instead of, "However, overall, experiment and theory are in excellent agreement." The later sentence is not informative and is neither a necessary or sufficient condition for the theory to be usefully relevant.

We thank the referee for this suggestion and modified the text accordingly.

Both of the below sentences need to be clarified. I do not understand the first and while the second is clearer I am not sure why no experimental signal is possible:

Thus, the probability density describing the position of the H₂ centre as function of the distance to the Cu site is highly justified. The Raman shift of the intramolecular frequency of H₂ varies strongly (several 100cm⁻¹, see Figure 4b) as function of R. As the adsorbed H₂ molecule is strongly delocalized (cf. the probability density in Figure 4b) a strongly shifted ν_R signal indicating the activated hydrogen molecule would be so broad that it cannot be detected in experiment"

We agree with the referee and have reworked this part of the manuscript. First, we denote the two important vibrational modes characterizing adsorbed H₂ as ν_R (for the mode describing the vibration of the H₂ centre of mass with respect to the adsorbing Cu(I) site, formerly ν_{IR}), and ν_I for the mode describing the intramolecular vibration (formerly ν_R to indicate Raman activity).

The reworked text reads as

“Thus, the calculated probability density describing the position of the H₂ centre as function of the distance to the Cu site is consistent with the difference in H₂ and D₂ adsorption enthalpy obtained from our experiments. This has direct implication on the intramolecular vibration (ν_r): In the interval of significant probability density $|\Psi(R)|^2$ the calculated ν_r value varies by several 100cm⁻¹ (Figure 4b). Thus, a strongly shifted ν_r signal, characterising the chemically activated hydrogen molecule, would be so broad that it cannot be detected in the Raman experiment.”

I also think a brief, simple explanation of how the zero point energies change the isotopic binding should be included - this paper is for a broader audience. Such a simple explanation would serve to inform the interesting nature of the results and focus readers on this simple quantum phenomenon (physical scientists may even forget the Born-Oppenheimer surfaces are not mass dependent).

We agree with the referee and reminded the reader at the point where we calculate the corresponding frequencies. The text reads now as:

“This allows capturing quantum-mechanically the isotope effect by analytical solution of the Schrödinger equation for the two vibrations associated with the adsorbed hydrogen isotopologues, resulting in frequencies ν_R and ν_r , associated zero point energies

$$E_0 = \frac{1}{2} h\nu - \frac{\left(\frac{1}{2} h\nu\right)^2}{4D}, \text{ and in the probability densities describing both vibrational modes}$$

(Table 1, Figure 4).”

The colors not closely matching the legend in Figure 1 needs to be addressed. It is a critical figure and it makes it difficult to interpret.

This has been corrected.

In the supplemental information the authors should also show the data in Figure S2 overlapped in a separate panel. The authors should also approximate the isosteric heat of sorption directly from the temperature dependent sorption data using Clausius-Clapeyron analysis. This is in a sense less model dependent and serves as a consistency check.

We thank the referee for this comment. In fact, we have now 4 ways to determine the adsorption enthalpy, all of them consistent with each other, and providing important insight. We used a van't Hoff plot (as suggested by the referee) to determine the isosteric heat of adsorption from the adsorption isotherms. The result is almost identical to that of the fitted Langmuir model, which justifies the single site first principles calculations,

which again agree with the experimental data. Finally, we report now TDS data which agree very well for D₂, but due to low adsorption density, a quantitative result for H₂ was not obtained.

In Figure S4 the caption line "The modified vibration occurs due to the reduced degrees of freedom of the physisorbed H₂ and D₂." is unclear. Do they mean the vibrational frequency shift occurs due to the attractive sorbate /sorbent interaction?

Yes, the referee's interpretation is correct. For further clarification we changed and extended figure caption S8 (former S4).

There are other language issues, even the first sentence:

"The metal-organic framework Cu(I)-MFU-4l captures heavy hydrogen isotopes deuterium and tritium from hydrogen gas phase mixture with D₂/H₂ separation factor of 11 at 100 K." isn't standard English. The text needs to be cleaned up for a broad audience in a high profile journal.

We reworded the abstract, also separated experimental results (involving D₂ and H₂) and theory (addressing also isotopologues involving T)

In this sentence: "The Born-Oppenheimer first principles calculations show a strongly anisotropic potential energy surface, both for the distance of the H₂ centre to the Cu(I) site (denoted as R) as well as for the bond length of the adsorbed H₂ species (denoted as r, see Figure 1)." The language here is not very clear although I think the intent is to refer to mapping the potential energy surface for the two coordinates defined in Figure 1 and they find an anisotropic surface of some sort. It is unclear what anisotropic means for a one dimensional coordinate. Clarity is required here. Is it exceptional anharmonicity? If so compared to what?

This was a typo, the PES is strongly anharmonic, i.e. the harmonic approximation fails for high-quality prediction of ZPE and vibrational frequencies.

I am not sure what this means: "Thus, the probability density describing the position of the H₂ centre as function of the distance to the Cu site is highly justified."

The purpose was to justify the calculations by showing their consistency with experiment. As discussed in an earlier comment by Referee 2 we have rephrased this part, confident that it is now clearly understandable.

The lack of clear, precise language throughout detracts from the important science.

In sum, the manuscript has many nice features. For it to be appropriate for a wide audience a clear compelling case needs to be made that this particular MOF has important distinct features from those previously characterized. None of the work is exceptional on its own - the effect is known in similar materials and the characterization techniques are used routinely. It is the combination of insightful experiments and theory that make the paper interesting. For it to be publishable a more polished manuscript with crisp, clear arguments and insights needs to be presented. Note, in my opinion, I have little issue with the data and the presentation of the results is adequate if not exceptional.

Reviewer #3 (Remarks to the Author):

This paper reports the captures of heavy hydrogen isotopes on MOF Cu(I)-MFU-4l. The experimental evidence is restricted to H₂ and D₂. The H₂/D₂ separation factor was 11 at 100K.

1) The hydrogen adsorption enthalpy is reported as 32 kJ.mol⁻¹. This value is very high, if not the highest value obtained so far and the standard deviation for the value should also be reported. As confirmation of this very high value, the authors should report an independent experimental measurement of the adsorption enthalpy for D₂, which should be slightly higher or the same as H₂, within experimental uncertainty. It is very important that such a high value for adsorption enthalpy is supported by additional experimental data.

We agree with the referee and have now four ways addressing the adsorption enthalpies both for H₂ and D₂, all of them consistent with each other, and with error bars addressed (see Tables S2-S5).

2) Applications of first principles calculations have been used to predict performance for the capture of tritium. However, the paper does not give experimental evidence to support the predictions. Therefore, I recommend a change in title for the paper to make it clear that the experimental results are restricted to H₂ and D₂.

On one hand we disagree with the referee: Our calculations have been validated by experiments involving H₂ and D₂. The isotope effect is solely reflected by the mass difference between the isotopes, so our predictions for isotopologues involving T are backed up by experiment, admittedly, only indirectly. We, however, agree with the referee that the wording of our text was misleading, and we have now clearly stated, also in title, abstract and introductory paragraph, what was directly measured, and which results are obtained by theory.

3) A significant amount of literature is available on isotope effects. Thermal desorption spectra as shown in Fig 2 have been reported previously by some of the authors of this paper - see Ref 16 etc. Therefore the most interesting aspect of this paper is the very high enthalpy obtained for MOF Cu(I)-MFU-4l and the explanation for potential selectivity.

We hope that we pointed out now clearly that the most interesting results are both the high selectivity at high temperature (which is expected due to the high adsorption enthalpy), and the isotope exchange mechanism that was not observed earlier (not even in a different material) to the best of our knowledge.

4) Page 8- last 4 lines. Comparison between INS and first principles calculations suggest a slight underestimation of R in the latter, which is consistent with a larger barrier. This can be due to the relatively small cluster model, which neglects long-range crystal interactions.

Please quantify the errors and the likely limitations for model predictions for T2 to increase confidence in the accuracy of the calculations since all models have approximations.

We have now performed high level-low level calculations (PBE-D3 in a periodic box at Gamma point approximation using plane waves, substantiated with PBE0-D3 calculations at the adsorption site in a subtraction scheme). The interaction energy slightly decreases (0.6 kJ.mol^{-1} due to long-range interactions from the other pore site and other ligands), but essentially the potential is only slightly shifted to higher values, keeping all other conclusions and predictions the same. In SI, we have added a paragraph discussing all potential shortcomings of the model and calculations (the most important ones, in this order, are probably (i) coupling of vibrational modes, (ii) neglect of cluster dynamics (phonons), (iii) intrinsic errors in DFT and quasi-classical treatment of London dispersion, (iv) intermolecular effects of adsorbed hydrogen, (v) the model of the strong adsorption site.).

5)Page 9 line 24

Interestingly, no isotope exchange to form mixed isotopologue HD has been observed, even though INS and first principle calculations confirm highly activated hydrogen at the Cu sites.

This appears to be a contradiction for the model, which requires further explanation.

We have addressed this point in the revised manuscript. The barrier associated with desorption is about 13 times lower than the barrier associated to dissociation (of the activated H₂), which is the rate determining step in intermolecular isotope exchange. All of our experiments have been performed at 200K or mostly even at lower temperatures and the respective time scale was below 1h. Therefore, no formation of HD is expected in significant quantities.

Reviewers' Comments:

Reviewer #2 (Remarks to the Author)

I believe the authors took the advice of the reviewers and proceed a clean clear manuscript. The significance of the material is now presented lucidly in the introduction along with an appropriate roadmap to the experimental evidence provided later.

I have no objections to the publication of the manuscript in its present form.

Reviewer #3 (Remarks to the Author)

The authors have attempted to address the comments of the reviewers and the paper has been improved. Comparable results have been reported in similar materials and the characterization techniques are not new and are used routinely. Similar results for H₂ and D₂ have been reported on different materials. The material studied has a high enthalpy of adsorption and this has been confirmed by several methods. The predictions of T₂ separation are only indirectly supported by experiments on H₂/D₂ and therefore, are an extension of the results without support of direct experimental data on T₂

Line 72 precise predictions for T₂ and mixed isotopologues

The word 'precise' should be deleted. The paper presents predictions without support of direct experimental data on T₂. The accuracy of the prediction can only be confirmed by direct experimental measurements on T₂.

Line 119 I think the error bar ± 0.1 kJ/mol quoted in the revised manuscript are optimistic!

Line 237 A very important result of this work is the strong isotope exchange mechanism.

I would not describe it as very important!

Line 242 H₂ is pushed out by slower

Replace 'pushed out' by 'displaced'.

Response to Reviewers:

Reviewer #3 (Remarks to the Author):

The authors have attempted to address the comments of the reviewers and the paper has been improved. Comparable results have been reported in similar materials and the characterization techniques are not new and are used routinely. Similar results for H₂ and D₂ have been reported on different materials. The material studied has a high enthalpy of adsorption and this has been confirmed by several methods. The predictions of T₂ separation are only indirectly supported by experiments on H₂/D₂ and therefore, are an extension of the results without support of direct experimental data on T₂.

We thank the referee for his remarks.

Line 72 precise predictions for T₂ and mixed isotopologues

The word 'precise' should be deleted. The paper presents predictions without support of direct experimental data on T₂. The accuracy of the prediction can only be confirmed by direct experimental measurements on T₂.

We do not completely share the opinion of the reviewer. The only difference in adsorption between hydrogen isotopologues are their nuclear mass and their nuclear spin. The latter effect is typically orders of magnitude smaller than the former one, which dominates the difference in adsorption enthalpies. Thus, if theory agrees well with all available data of two isotopologues, including adsorption enthalpy and geometry, we can confidently make predictions to other isotopologues. This is particularly important for the present study, as working with tritium in gas phase requires severe security measures that are only available in few dedicated laboratories. Thus, our approach is example for responsible research where hazardous procedures are avoided whenever possible.

Line 119 I think the error bar ± 0.1 kJ/mol quoted in the revised manuscript are optimistic!

This error bar addresses the mathematical procedure in generating the data from the experimental isotherms and do not include systematic errors in the measurements. We have mentioned this now explicitly in the text.

Line 237 A very important result of this work is the strong isotope exchange mechanism. I would not describe it as very important!

We disagree with the referee, and believe that we did not make this point clear enough in the manuscript: A strongly adsorbing site leads to kinetically controlled adsorption mechanism if molecular exchange is impossible. This is the fact here at temperatures below 80K. At higher temperature, the process is thermodynamically controlled and more suitable for hydrogen isotopologue separation. We have mentioned this now explicitly in the abstract and in the discussion.

Line 242 H₂ is pushed out by slower

Replace 'pushed out' by 'displaced'.

We agree that "pushed out" is not appropriate language, however, we believe that replaced is the correct term.